# Dietary Support in Elderly Patients with Inflammatory Bowel Disease

**DOI:** 10.3390/nu11061421

**Published:** 2019-06-24

**Authors:** Piotr Eder, Alina Niezgódka, Iwona Krela-Kaźmierczak, Kamila Stawczyk-Eder, Estera Banasik, Agnieszka Dobrowolska

**Affiliations:** Department of Gastroenterology, Dietetics and Internal Medicine, Poznan University of Medical Sciences, Heliodor Święcicki Hospital, 60-355 Poznań, Poland; niezgodkaalina@gmail.com (A.N.); krela@op.pl (I.K.-K.); kamilastawczyk@wp.pl (K.S.-E.); esta717@gmail.com (E.B.); agdob@ump.edu.pl (A.D.)

**Keywords:** inflammatory bowel disease, malnutrition, Mediterranean diet, older age

## Abstract

Ageing of the human population has become a big challenge for health care systems worldwide. On the other hand, the number of elderly patients with inflammatory bowel disease (IBD) is also increasing. Considering the unique clinical characteristics of this subpopulation, including many comorbidities and polypharmacy, the current therapeutic guidelines for the management of IBD should be individualized and applied with caution. This is why the role of non-pharmacological treatments is of special significance. Since both IBD and older age are independent risk factors of nutritional deficiencies, appropriate dietary support should be an important part of the therapeutic approach. In this review paper we discuss the interrelations between IBD, older age, and malnutrition. We also present the current knowledge on the utility of different diets in the management of IBD. Considering the limited data on how to support IBD therapy by nutritional intervention, we focus on the Mediterranean and Dietary Approaches to Stop Hypertension diets, which seem to be the most beneficial in this patient group. We also discuss some new findings on their hypothetical anti-inflammatory influence on the course of IBD.

## 1. Introduction

The frequency of Crohn’s disease (CD) and ulcerative colitis (UC), two forms of inflammatory bowel disease (IBD), is increasing worldwide [1,2,3]. Simultaneously with the demographic ageing of the human population observed in recent decades, especially in developed countries, the number of elderly IBD patients is also increasing [4,5,6,7,8]. Considering the definition of elderly as aged 60 years and above, it is estimated that 25–35% of CD and UC patients meet this criterion. These data encompass both those who were diagnosed before reaching 60 and those who were diagnosed when over 60 (elderly onset). The latter group, representing 10–15% of all IBD patients, reflects the second peak of CD and UC morbidity [9]. UC is the more frequent IBD subtype in this age range, since one in eight UC patients is older than 60, compared to one in 20 CD cases [9]. A population-based cohort study by Charpentier et al. revealed that, among elderly people suffering from IBD, 65% are between 60 and 70 years old, 25% between 70 and 80 years old, and 10% are older than 80 [10].

There is an increasing body of evidence showing several differences in the clinical course and management of elderly IBD patients, compared with those suffering from UC or CD at a younger age. One of the most important characteristics is a tendency for less aggressive therapeutic regimens [11,12,13]. On the other hand, the significance of non-pharmacological and non-surgical interventions is higher. Since both IBD and older age are independent risk factors of nutritional deficiencies, appropriate dietary support is needed, especially in the cases of patients older than 60 with UC or CD [12,14,15]. In this review, we discuss the influence of older age on the physiology of the gastrointestinal tract and the mechanisms leading to malnutrition, especially in the context of IBD in the elderly. We summarize the main differences in the clinical course and management of IBD in the elderly with special emphasis on the role of diet. We also present some recommendations for nutritional support for this unique population.

In order to analyze the current literature on dietary support among elderly IBD patients, we searched the PubMed and Web of Science databases using the key words “diet and IBD”, “elderly IBD”, “diet in elderly people”, and “nutrition in elderly IBD patients”. We identified mainly review papers, meta-analyses, and guidelines published after 2010. Moreover, we analyzed conference abstracts from the Congresses of the European Crohn’s and Colitis Organisation (ECCO) in 2018 and 2019.

### 1.1. Elderly IBD Patient—Differences in Clinical Course and Management

There are several characteristic features of elderly IBD. The delay in the diagnosis of CD or UC is longer than that in younger patients. This is related to less specific symptoms and to a frequent co-existence of other comorbidities and polypharmacy. Differential diagnosis encompasses many entities like ischemic colitis, infectious diseases, drug adverse reactions (non-steroidal anti-inflammatory drugs, anticoagulation, anti-platelet drugs, chemotherapy, etc.), diverticulitis, radiation colitis, and microscopic colitis. Performing invasive diagnostic investigations (like colonoscopy) can also be challenging, since older age, other comorbidities and drugs used are risk factors for severe complications [5,6,16,17].

The clinical course of elderly IBD seems to be less aggressive [6,8]. In CD, there is a higher frequency of colonic location [16,18]. On the other hand, complications like strictures or perianal involvement and extraintestinal manifestations are less common. As a result, the clinical presentation of elderly CD can be similar to UC, with rectal bleeding as a main symptom. Abdominal pain, weight loss, and diarrhea are less typical [16,18,19,20]. In the case of elderly UC, the predominant location is E2 or E3 according to the Montreal classification, whereas isolated proctitis is rare [20,21]. The disease is more stable over time and there is a low frequency of proximal disease colonic extension [19,20]. The need for a colectomy is also relatively low. A French population-based registry (EPIMAD) showed that only 16% of elderly onset UC patients underwent a colectomy in a ten-year follow-up period [6,8,22,23,24].

Despite a milder clinical course in long-term observation, the first IBD episode can be paradoxically more severe than in younger patients [18]. Older age also seems to be related to a more frequent hospitalization rate in IBD [18,20]. A study by Ananthakrishnan et al. revealed that hospitalized IBD patients older than 65 are at a higher risk of significant malnutrition, anaemia, and hypovolemia [25]. The frequency of thromboembolic complications is increased due to hypercoagulability, dehydration, prolonged bed rest, and immobilization. This is why the hospitalization of elderly IBD patients seems to be connected with higher fatality [18,26,27,28,29].

Although there are no randomized, controlled trials assessing the therapeutic strategies for IBD in the elderly, medical and surgical management is often different from patients of a younger age [30,31,32,33,34]. The usage of many medications is limited due to their higher toxicity (e.g., corticosteroids), the risk of interactions (e.g., thiopurines with allopurinol, mesalamine with anticoagulants), contraindications (e.g., renal insufficiency in the case of mesalamine, severe congestive heart failure in the case of anti-tumor necrosis factor alpha antibodies), and higher rates of adverse events (e.g., serious infections, diabetes, arterial hypertension, mental disorders in the case of corticosteroids or neoplastic complications in the case of thiopurines) [7,35,36,37,38,39,40,41,42,43,44,45,46,47,48,49,50,51,52,53]. The safety of newly registered immunosuppressive molecules and biological agents (tofacitinib, anti-integrins–vedolizumab, or anti-IL-12/23 antibodies–ustekinumab) in elderly IBD patients has not been studied at all [54].

General indications for surgery in IBD patients aged >60 are similar, when compared with the younger subgroup, however, a decision to use surgical intervention should be taken with caution since there is a higher risk of post-operative complications and mortality [5,11]. Nevertheless, in many cases surgery is inevitable, which is why, in order to improve therapeutic outcomes, optimal treatment should be applied preoperatively, with minimization of corticosteroid use and extensive nutritional support [5,11,55,56,57].

The rules for disease monitoring in elderly IBD patients should be also adjusted for age and concomitant morbidities. Since repeated endoscopic assessment is often impossible, the importance of non-invasive markers of inflammatory activity, like fecal calprotectin or C-reactive protein, is high [58,59]. In terms of cross-sectional imaging methods, repeated computed tomography or magnetic resonance (MR) imaging can be difficult and, in many cases, contraindicated [60]. This is due to the fact that a significant proportion of older patients suffer from renal insufficiency or are at a high risk of this complication, which makes the administration of an intravenous contrast agent impossible. Another limitation for MR imaging is the high frequency of metallic implants (e.g., after a total hip or knee replacement or after the implantation of a cardiac rhythm control device) in elderly people. This is why more common use of an abdominal ultrasound should be advised for the objective assessment of morphological abnormalities in the gastrointestinal tract [60].

### 1.2. Ageing, IBD, and Malnutrition—What Are the Connections?

As discussed above, there are many limitations for the routine application of classical therapeutic approaches in the case of IBD in elderly patients. Thus, non-pharmacological and non-surgical interventions are of great importance. The role of dietary support is especially high, since ageing by itself increases the risk of malnutrition. Epidemiological analyses show that 5%–20% of European citizens aged 60 and older suffer from malnutrition, while for hospitalized patients or those in long-term care, these numbers are even higher [61]. Thus, obligatory assessment of the nutritional status of all older patients is recommended by both the American Society for Parenteral and Enteral Nutrition (ASPEN) and the European Society for Parenteral and Enteral Nutrition (ESPEN). A Mini Nutritional Assessment (MNA) is believed to be the most appropriate tool for this purpose [62,63,64].

The etiology of malnutrition in older people is multifactorial. There are multiple medical conditions associated with a high risk of weight loss and nutritional deficiencies, like cancer, pulmonary disorders (chronic obstructive pulmonary disease), diabetes, cerebrovascular and neurological diseases, and gastrointestinal disorders. Many of those conditions are characterized by an increased catabolism, loss of appetite, and dysphagia. Multimorbidity and polypharmacy—typical phenomena among elderly people—are also connected with higher hospitalization rates, and increased probability of significant drug interactions [65]. These factors can independently promote malnutrition [65,66,67]. Another important problem is poor oral health and dental status leading to chewing difficulties and mouth dryness, which can cause lower food intake [68,69]. Depression, anxiety, dementia, and many other neuropsychological factors can result in unintentional weight loss and nutritional deficiencies [70,71,72]. There are also many social determinants of malnutrition risk, like poverty, loneliness and isolation, an inability to shop or cook, secondary to cognitive disorders and/or physical disability [73,74]. Interestingly, although ageing per se is not always associated with malnutrition, there are several physiological phenomena increasing the risk of weight loss. Decreasing appetite among elderly and otherwise healthy people can be explained by a reduction in stomach capacity and impairment of gastric relaxation, accompanied by lower gastric emptying. One of the etiological hypotheses for these processes in older people is fluctuation in the production and secretion of several enterohormones. There are data suggesting that higher levels of cholecystokinin and lower concentration of ghrelin can contribute to early satiation after food consumption. Moreover, degenerative processes in the gastrointestinal tract can result in the reduction in the number of taste buds, which can be accompanied by a deterioration in the sense of smell. These phenomena can additionally demotivate the patients to consume regularly [75,76,77].

Since the mechanisms underlying malnutrition in elderly people are complex, three types of weight loss proposed by Roubenoff can coexist: wasting, cachexia, and sarcopenia [78,79]. Wasting is associated with inadequate dietary intake and results in involuntary weight loss [80]. Cachexia is caused by induced catabolic processes, with pro-inflammatory cytokines like interleukin-1 (IL-1), tumor necrosis factor–alpha (TNF-alpha), IL-6, and others having a predominant role. The main consequence of cachexia is a decrease in fat-free mass and body cell mass [81]. Sarcopenia is defined as a loss of muscle mass [78,79,82]. The etiology of this phenomenon is poorly understood, however, a dominant role is hypothetically played by a lack of physical activity, induction of the pro-inflammatory response, and dysregulation of anabolic hormones, like testosterone or growth hormone [83,84].

IBD is independently associated with an increased risk of malnutrition. Epidemiological data shows that 65%–75% of CD patients and 18%–62% of UC patients have nutritional deficiencies [85]. The discrepancies in these numbers are a consequence of the different definitions of malnutrition. Body mass index (BMI) is among the most frequently used criteria, but it has been widely criticized recently, since it does not take into account several qualitative and quantitative parameters like the relation between fat and muscle mass, the concentration of micro- and macronutrients, recent changes in body mass or disease activity [86]. There are many data showing that low body mass is only one of the dimensions reflecting malnutrition in IBD. Among other parameters, which have been frequently reported, are deficiencies in iron, calcium, selenium, vitamin D and/or vitamin K [85].

The etiology of malnutrition in IBD is complex. It encompasses disease-related and treatment-related factors [85]. In the first group, a decrease in food intake seems to be the most important. This phenomenon can be related to IBD symptoms, like nausea, vomiting, abdominal pain, diarrhea, fever or fatigue. Also, it has been shown that hospitalization is associated with a higher risk of inappropriate food intake due to a frequent need to fast in preparation for different investigations or due to an inadequate hospital diet [85,87]. Moreover, disease activity by itself, with the production of multiple pro-inflammatory cytokines, induces catabolic processes, and promotes increased energy expenditure, contributing to malnutrition. The absorptive functions of the gastrointestinal tract are also impaired due to bowel wall damage with a loss of epithelial integrity, bacterial overgrowth, and increased intestinal motility. The same factors contribute to enhanced nutrient loss [85,88]. Considering the treatment-related causes of malnutrition, there are data on the negative impact of steroids on body composition. Moreover, nitroimidazoles or immunosuppressive drugs (thiopurines, methotrexate) can also alter the appetite, leading to reduced food intake [85]. On the other hand, multiple surgical resections limit the absorptive gastrointestinal surface, in spite of the high compensatory potential of the remaining parts of the intestines [85].

Taking into consideration the complex etiology and high frequency of malnutrition among elderly people and IBD patients analyzed separately, the significance of this phenomenon among CD and UC patients aged 60 and older becomes especially challenging. In order to prevent and/or adequately treat this unique subpopulation, proper dietary support is needed.

### 1.3. The Role of Diet in IBD and the Elderly

There are no strict dietitian recommendations for patients with IBD, since there are insufficient data for promoting any special diet. The general rule is that patients should cover their energy demand by eating well-balanced meals containing complex carbohydrates, proteins and fats, mainly of plant origin, rich in vegetables and fruits, with the elimination of highly processed foods [89]. Special attention should be paid to appropriate iron and vitamin D consumption. In each case, however, a highly individualized recommendation should be defined in order to adjust the nutritional needs to a concrete, clinical scenario, especially in older people.

According to current knowledge, in the case of physiological ageing special recommendations should be given for protein, vitamin D, and water consumption. In order to maintain muscle mass, the PROT-AGE study group defined the daily protein requirement, which is 1.0–1.2 g protein/kg body weight [90]. Moreover, all individuals should supplement vitamin D3 in a dose of 800–2000 IU per day [91]. Adults should drink 30–35 mL/kg body weight (at least 1500 mL/day or 1–1.5 mL/1 kcal) of water (preferably medium-carbonized and still water). Since there is an increased risk of dehydration among older people, the recommendations for daily water consumption in this subpopulation (similar to the pediatric population) are more precisely defined as 100 mL of water for the first 10 kg, then 50 mL for second 10 kg, and 15–20 mL for each additional kilogram of body weight [92,93]. In addition, older adults are in the groups at risk of vitamin B12 deficiency. The usual dietary sources of vitamin B12 are animal products, including fish, meat, poultry, eggs, milk, and milk products. Vitamin B12 is generally not present in plant foods, but a lot of these products are fortified. The vitamin B12 recommended dietary allowance for older adults is 2.4 μg/day [94,95].

In the case of high IBD activity, especially in patients with severe diarrhea and abdominal pain due to stricturing CD, it is advisable to avoid a high intake of fiber and lactose, in order to prevent bacterial overgrowth and reduce the number of bowel movements [96]. The daily protein requirement is 1.2–1.5 g protein/kg body weight [96]. Resting energy expenditure during a flare is 25–30 kcal/kg standard body weight [96,97,98]. Moreover, according to the ESPEN recommendations, oral nutrition supplements (ONS) should be considered in addition to a normal diet for the treatment of nutritional deficiencies in the case of IBD exacerbation [96]. ONS contain high amounts of all (complete) or selected (incomplete) macro- and microelements in relatively small volume products. ONS can be also divided into two categories: standard ONS which contain different nutritional compounds in proportions characteristic for a normal oral diet and specific ONS which is composed adequately for some particular patient populations (e.g., Parkinson’s disease, Alzheimer’s disease, etc.). ONS can be used together with meals; they contain no lactose, gluten, purines or cholesterol and should be considered in each case of increased malnutrition risk or diagnosed malnutrition [96,99].

Recent years, however, have brought plenty of data about the crucial role of impaired microbiota in the pathogenesis of intestinal inflammation. Moreover, there is a growing body of evidence that also ageing is associated with changes in intestinal microbiota composition. In 2007 the ELDERMET consortium was established to investigate this topic [100]. They found (by using the pyrosequencing of 16S rRNA method) that there was an increase in *Bacteroidetes* and a concomitant decrease in *Firmicutes* species among older people, however there was a significant inter-individual variability in the composition of elderly gut microbiota [101]. One of the reasons was the health status of the investigated subjects. The statistical analysis indicated a clear separation between community-dwelling subjects and long-stay home residents [101]. Another observation was that health status and the diversity of the intestinal microbiota in the ELDERMET study correlated with the patients’ nutritional habits. It was shown that the diversity index of the fecal microbiota was significantly associated with a low-fat and high-fiber diet [101]. It is, however, still not known whether changes in gut microbiota are a result of dietary intervention or are more related to unhealthy ageing by itself.

Nevertheless, the possibility of shaping the intestinal microbiota by nutritional interventions would be very attractive. The hypothetical promotion of a “healthy” in-vironment (microbiota) by environmental factors (diet) seems to be an interesting concept for therapeutic intervention also in IBD. This is why dietary intervention is currently considered not only in the context of sufficient nutritional support, but also as a potential modulator of intestinal inflammation [102,103]. Our understanding of the link between nutrition, intestinal microbiota, and inflammatory response is still poor, however, due to the development of new technologies such as metabolic profiling and next-generation DNA sequencing, we know that microbiota composition changes after exposure to different modifying factors [104]. For example, there are data showing that a high-fat and low-fiber diet, as well as an animal-based diet, increase the abundance of *Bacteroidetes* and *Prevotella*, which are believed to participate in the development of chronic inflammation in the gastrointestinal tract [105]. On the other hand, dietary fiber can promote short-chain fatty acids synthesis by colonic microbiota, which can lead to the suppression of pro-inflammatory cytokines from dendritic cells and macrophages [104,106,107]. Another hypothetical association between the diet and inflammation is the epigenetic regulation of gene expression by different nutritional components. There is some evidence that the typical Western diet, deficient in micronutrients, like selenium and folate, can influence DNA methylation, which promotes pro-inflammatory phenomena and seems to increase colorectal cancer susceptibility [104]. What is more, in an experimental model of IBD it was shown that selenium supplementation prevented tissue damage through interfering with the expression of the key genes responsible for inflammation [104,108]. Nevertheless, although these concepts of the associations between diet, microbiota, and inflammatory response are very promising, we are still not able to translate this knowledge into clinical practice. We hope that it will be possible in the future to modulate our microbiota by changing the in-vironmental milieu via nutritional intervention, but we still need more data.

Among different diets already studied in the context of IBD, main attention is being paid to the low-fermentable oligosaccharide, disaccharide, monosaccharide, and polyol (FODMAP) and anti-inflammatory diet (IBD-AID), although supporting scientific evidence is relatively poor [104]. Recently, the advantages of the Mediterranean or the Dietary Approach to Stop Hypertension (DASH) diets in the context of chronic inflammation have also been discussed [105,106].

The main rule of the low-FODMAP diet is to exclude highly fermentable and poorly absorbed carbohydrates and polyols. In this diet, consumption of different food types is strongly discouraged, such as many fruits (e.g., apple, blackberry, grapefruit, mango, nectarine, peach, plum, watermelon), vegetables (e.g., artichoke, asparagus, avocado, onion, cabbage, garlic, leek, pea), dairy (e.g., cow, goat, sheep, condensed and evaporated milk), beverages (e.g., green tea, soft drinks, white tea, coconut water) and many nuts, seeds and legumes. Moreover, breaded meat or meat made with high fructose corn syrup should be avoided. This is not a long-term diet and the dietary limitations should last for only 6–8 weeks. Then patients should gradually restart foods high in FODMAPs in order to establish an individual tolerance to specific oligosaccharides, disaccharides, monosaccharides, and polyols. The utility of the low-FODMAP diet has been shown mainly for patients with irritable bowel syndrome (IBS), since there is a hypothesis that high fermentation in the gastrointestinal lumen can lead to increased intestinal permeability and provoke intestinal hypersensitivity in a genetically susceptible host [107,108]. Data on the usefulness of this diet in IBD are limited and mainly come from retrospective cohorts. It is advised that a low-FODMAP diet can be used in selected IBD patients with IBS-like symptoms in addition to conventional therapy, but only under strict dietitian supervision [104,105,106,107,108,109,110,111,112,113,114,115].

IBD-AID is a multistep and highly individualized dietary intervention, limiting some specific carbohydrates (e.g., refined sugar, gluten-based grains, certain starches). Olendzki et al., who developed IBD-AID, hypothesized that this can decrease the growth of several pro-inflammatory bacteria in the gastrointestinal tract, preventing dysbiosis [116]. In the next step, the patient should ingest prebiotics and probiotics (e.g., leek, onion, fermented food) to promote restoration of the microbiota. Moreover, the consumption of total and saturated fat, and hydrogenated oils should be avoided, together with the individual identification of dietary intolerances and nutritional deficiencies. The rules of IBD-AID were first published in 2017 and until now there were no randomized, controlled trials conducted in order to confirm the initial, promising reports on the use of this diet as an adjunct therapy for the treatment of IBD [104,116].

Lack of sufficient data for the usefulness of the low-FODMAP diet and IBD-AID in IBD result in a high skepticism of clinicians to promote this kind of dietary intervention. In the case of elderly IBD patients, another important limitation for the use of these diets is their complexity. Moreover, there is a high risk of several nutritional deficiencies due to the restriction and avoidance of different foods, especially when the dietary intervention is conducted without professional support. This can have serious negative consequences, considering the general increased risk of malnutrition and the presence of serious comorbidities in older people. This is why it seems to be more reasonable to promote safer diets, with more robust data in the context of elderly patients.

Considering the nutritional requirements and characteristics of elderly patients with IBD discussed above, as well as the most common disorders among older people (arterial hypertension and other cardiovascular diseases, type 2 diabetes, hypercholesterolemia), the DASH or Mediterranean diet could be recommended for this unique population. The main restrictions in the DASH diet concern carbohydrates and fats, in particular by limiting simple carbohydrates (glucose, fructose, saccharose) and reducing the intake of saturated fats. This means a significant reduction in the consumption of sweets, sugar confectionery, sweeteners, fruit preservatives (less than five portions per week), as well as red meat and highly processed food. Vegetables and fruits should be eaten 4–5 times/day, and whole grain products 6–8 times/day. The DASH diet also includes medium-fat dairy products (2–3 portions/day), however, this needs to be accompanied with regular consumption of vegetable oils (preferably raw, inter alia, to enable the absorption of fat-soluble vitamins). The recommended frequency for eating fatty saltwater fish (herring, salmon, mackerel, halibut, sardine, codfish, flounder) is 2–4 times/week. Different seeds, nuts, legumes are also an important part of the DASH diet, since they contain (similar to vegetable oils—linseed, soybean or rapeseed oil— and fatty saltwater fish) high amounts of omega-3 polyunsaturated fatty acids (PUFA). Another main rule of this type of diet is a significant reduction in salt (sodium) consumption [117].

Although there are no data on the utility of the DASH diet in IBD, its beneficial effect on general health status and cardiovascular risk is well known [118,119,120,121]. Moreover, Nilsson et al. showed that adherence to a DASH-style diet was significantly associated with a lower clustered metabolic risk among older women, and it promoted a systemic anti-inflammatory environment, independently of physical activity [122]. The authors concluded that the DASH diet should be considered as a key target for nutritional intervention among elderly people to prevent age-related metabolic abnormalities.

The general recommendations of the Mediterranean diet are very similar to DASH. It emphasizes eating primarily plant-based foods (fruits, vegetables, whole grains, legumes, nuts, seeds, olive oil), which should be consumed several times per day, together with dairy products (mainly different types of cheese, yogurts). Low or moderate alcohol drinking (preferably red wine with a meal) is also advised. Fish, eggs, and poultry can be consumed several times per week. In contrast, consumption of sweets and red meat should be significantly reduced (a few times per month). The details of the Mediterranean diet can vary depending on the region of the Mediterranean Basin; however, the general rule is to eat foods coming from this geographic area [123,124].

In contrast to the DASH diet, there are some data on the usefulness of Mediterranean diet in IBD. Marlow et al. demonstrated that even a short-term (six weeks) nutritional intervention is beneficial for patients with CD, decreasing the concentration of several pro-inflammatory markers with a trend to normalize the composition of intestinal microbiota. Transcriptomics analyses confirmed small changes in many genes, providing a cumulative anti-inflammatory effect of the diet [125]. In another study, Godny and colleagues showed that the Mediterranean diet is associated with decreased fecal calprotectin in patients after pouch surgery in UC, which is accompanied by an improvement in gut microbiota composition [126]. Moreover, Molendijk et al. demonstrated a beneficial effect of long-term nutritional intervention in IBD. In this study, six months of the Mediterranean diet improved the quality of life and reduced CRP levels. The level of improvement was associated with adherence to the rules of this type of diet [127].

The question remains, which hypothetical mechanisms could be related to the anti-inflammatory properties of the Mediterranean diet in IBD. As discussed above, this diet is characterized by a low intake of omega-6 PUFA, high intake of omega-3 PUFA, and dietary fiber, which seems to be important in the context of IBD. In line with that, the most recent epidemiological data indicate that a higher ratio of omega-6/omega-3 PUFA in the diet can be associated with an increased UC incidence [128]. Moreover, Hou et al. noted that a high intake of omega-6 PUFA, saturated fats, and meat is correlated with an increased risk of developing UC and CD [129]. It was also shown in a murine dextran sulfate sodium (DSS)-induced colitis model that omega-6/omega-3 PUFA ratio in the diet can influence the inflammatory processes in the gastrointestinal tract [130]. The authors observed that an α-linolenic acid (ALA)-enriched diet with a decreased uptake of linoleic acid (LA) resulted in less severe colitis in mice, with a markedly alleviated intestinal inflammation [130]. This was supported by Pearl et al. who showed the association between severity of intestinal inflammation and increased content of omega-6 PUFA in inflamed mucosa in UC patients [131]. Furthermore, Uchiyama et al. investigated the influence of a diet therapy involving the use of an “omega-3 PUFA food exchange table”. The authors showed that omega-3 PUFA significantly increased the erythrocyte membrane omega-3/omega-6 PUFA ratio in IBD patients, what was associated with clinical remission of the disease [132]. Recently, another experimental study on the protective role of omega-3 PUFA has been published. Charpentier et al. showed that supplementation of omega-3 PUFA significantly decreased colon inducible nitric oxide synthase (iNOS) and cyclooxygenase-2 (COX-2) expression, as well as IL-6 and leukotriene B4 production in 2,4,6-trinitrobenzene sulfonic acid (TNBS)-induced colitis [133].

Dietary fiber is also believed to have a protective effect on the development of inflammation in the gastrointestinal tract. Moreover, the short-chain fatty acids, regarded as one of the major microbial metabolites of dietary fiber, have the potential to improve intestinal mucosal immunity and maintain homeostasis [134]. There are several experimental and clinical data supporting these hypotheses. Liu et al. showed in a murine DSS-induced colitis model that supplementation of β-glucans at a dose of 500 mg/kg per day reduced the severity of clinical activity of the disease. β-glucans-enriched diet resulted in a smaller weight loss, improvement in the number of bowel movements, and amelioration of the inflammatory response assessed microscopically. It has been also shown that β-glucans supplementation inhibited the expression of pro-inflammatory proteins, such as TNF-α, IL-1, IL-6 or NOS [135]. On the other hand, based on the data from the Nurses’ Health Study, it was suggested in a prospective study that a long-term intake of dietary fiber was associated with lower risk of CD, but not UC [136]. A meta-analysis, performed by Liu and colleagues, indicated that the intake of dietary fiber was related to a decreased risk of developing IBD [137]. In a recent study by Andersen et al. an inverse association between the consumption of cereal fiber and CD in non-smokers was confirmed [138].

The only theoretical limitation of the DASH or Mediterranean diet in IBD is the high amount of whole grain cereal products, nuts, and seeds of leguminous plants which can stimulate intestinal peristalsis and increase the frequency of bowel movements. This is why it is advised to reduce the consumption of these particular foods during an IBD flare, whereas in patients in remission the individually tolerated amount of these products should be established.

The main rules of DASH and the Mediterranean diet are presented in Table 1 and Table 2.

## 2. Conclusions

The ageing of the human population has become a big challenge for health care systems worldwide. The increasing proportion of elderly people is a result of significant improvements in medical care, successful prophylaxis of infectious diseases, and declining birth rates in developed countries. On the other hand, the number of elderly IBD patients is also increasing and we have to face the problem of managing this unique population. Since there are several important differences in the clinical characteristics of older IBD patients, appropriate nutritional intervention and counseling should become a crucial element of the therapy. Although there are no data on the definite therapeutic influence of any diet on the course of IBD, it seems to be reasonable, considering data presented in this paper, to actively promote a healthy diet among elderly patients with IBD with special emphasis on the DASH or Mediterranean-style diet. Patients with IBD aged >60 are also at increased risk of cardiovascular diseases, type 2 diabetes, and arterial hypertension. This is why these two similar types of nutrition can cover not only the dietary requirements characteristic of a chronic inflammatory condition, but also due to its anti-inflammatory properties, they can improve the metabolic abnormalities typical in older age. Of course, it seems rational to advocate these types of diets only in parallel with classical treatment of IBD and even regardless of subsequent gastrointestinal disorders or any other disease. Nevertheless, since application of the current, aggressive therapeutic approaches in a significant proportion of elderly IBD patients is limited, the use of the Mediterranean and DASH diets is reasonable, especially in this unique population. 

## Figures and Tables

**Table 1 nutrients-11-01421-t001:** The rules of Dietary Approaches to Stop Hypertension (DASH) diet [117].

The DASH Diet (Dietary Approaches to Stop Hypertension)			
Dietary Product	The Frequency of Consumption	Indicated	Contraindicated
Cereal products	6–8/day	whole grain	refined
Vegetables	4–5/day	all	-
Fruits	4–5/day	all	-
Protein	6 or less/day	fatty saltwater fish, lean meat, seeds of leguminous plants	fatty, red meat
Nuts and seeds	4–5/week	all	-
Fats	2–3/day	vegetable oils rich in unsaturated fatty acids	animal fat, coconut oil, palm oil
Dairy products	2–3/day	low-fat or fat-free	full-fat
Drinks	several times a day	unspecified	drinks containing simple carbohydrates
Other			
Sweets, confectionery products	5 or less/week	-	-
Sodium	Max. 2300 mg/day	-	-

**Table 2 nutrients-11-01421-t002:** The rules of the Mediterranean diet [124].

The Mediterranean Diet			
Dietary Product	The Frequency of Consumption	Indicated	Contraindicated
Cereal products	several times a day	whole grains	refined
Vegetables	several times a day	all	-
Fruits	several times a day	all	-
Fish and seafood	several times a week (at least 2 times a week)	fatty saltwater fish (tuna, salmon, sardines, herring) and mussels, oysters and shrimps	-
Poultry and eggs	several times a week	all	-
Red meat	a few times a month	-	-
Nuts and seeds of leguminous plants	several times a day	all	-
Fats	several times a day	olive oil	animal fats such as lard, butter, fatty beef, fatty pork, poultry with skin
Dairy products	several times a day	all	-
Drinks	several times a day	still water	sugary drinks
Sweets, confectionery products	few times a week	-	-
Red wine	every day; women max. 1, men max. 2 glasses/day	-	-

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
