# Peer review of "Dietary Support in Elderly Patients with Inflammatory Bowel Disease"

_nutrients, 2019, doi:10.3390/nu11061421_

Round 1
Reviewer 1 Report
The paper provides a good topic and is linked to some recent data regarding diet modifies in IBD patients;nevertheless it's not completely described and clarified the real gain of special diet behaviour in elderly compared with younger patients. Moreover, mediterranean and DASH diet showed a positive effect in all multimorbidity and polypharmacy, not only in IBD patients; it's not clear if these diets could be an exclusive helpful in intestinal disorder or they should be evocated in elderly population regardless of intestinal disease. Please, may the authors explain this point.
Author Response
Dear Reviewer,
I would like to thank you for giving us a chance to revise our paper. The paper has been checked and corrected thoroughly according to all the comments. All corrections are marked in red in the file containing revised version of our paper.
Once again, I would like to express my sincere gratitude for giving us the opportunity to revise and resubmit our paper.
Comment No 1: The paper provides a good topic and is linked to some recent data regarding diet modifies in IBD patients;nevertheless it's not completely described and clarified the real gain of special diet behaviour in elderly compared with younger patients. Moreover, mediterranean and DASH diet showed a positive effect in all multimorbidity and polypharmacy, not only in IBD patients; it's not clear if these diets could be an exclusive helpful in intestinal disorder or they should be evocated in elderly population regardless of intestinal disease. Please, may the authors explain this point.
Reply to Reviewer No 1: Dear Reviewer, we would like to thank you for this comment. As you mentioned, Mediterranean and DASH diet are not exclusively dedicated for elderly IBD patients. It seems rational to promote these diets among older people regardless of the subsequent GI disorders or any other diseases. What’s more – it can be advocated also for younger IBD patients. Nevertheless, we wanted to stress the fact that there are some new data on the utility and potential anti-inflammatory properties of these types of nutritional interventions in IBD patients. Taking into account the limited possibilities of an aggressive therapeutic approach in substantial proportion of elderly IBD patients, it seems reasonable to promote these type of diets in this unique population. Of course, we realize the fact that there are still no data on the definite therapeutic influence of any diet on IBD course. That is why we advocate Mediterranean or DASH diet only in parallel to classical therapy of IBD. We added some additional comments on that in the “Conclusions” section of our paper.

Reviewer 2 Report
Please have the paper reviewed for grammar - at this point the paper is very challenging to read.
Lines 56-88 - many statements need references
171-179 - what about vitamin B12?
Some of the recommendations for diet are very outdated in light of the current literature on the microbiome
Conclusions are supported by insufficient evidence
Author Response
Dear Reviewer,
I would like to thank you for giving us a chance to revise our paper. The paper has been checked and corrected thoroughly according to all the comments. All corrections are marked in red in the file containing revised version of our paper.
Once again, I would like to express my sincere gratitude for giving us the opportunity to revise and resubmit our paper.
Reviewer No 2
Comment No 1: Please have the paper reviewed for grammar - at this point the paper is very challenging to read.
Reply No 1: The paper was reviewed and corrected by a professional native-speaking person.
Comment No 2: Lines 56-88 - many statements need references
Reply No 2: References have been added.
Comment No 3: 171-179 - what about vitamin B12?
Reply No 3: Dear Reviewer, we would like to thank you for this comment. The capacity to absorb vitamin B12 from a food-based diet decreases in older adults and over time can result in the food-cobalamin malabsorption syndrome, characterized by mild vitamin B12 deficiency, decreased whole body stores and metabolic disturbances. What's more the vitamin B12 deficiency is more common in elderly patients with IBD. The vitamin B12 RDA for adults is 2.4 μg/day. We have added information about the vitamin B12 in our paper.
Comment No 4: Some of the recommendations for diet are very outdated in light of the current literature on the microbiome
Reply No 4: We agree that the knowledge on the role of the microbiota is increasing rapidly in recent years. However, we think that it is too early to recommend anything in terms of diet in the context of the current literature on the microbiome. The only known diet, which theoretically is related to the intestinal microbiota, is AID, which is discussed in the paper, although the scientific data on the influence of AID on microbiota is poor. According to the current theoretical knowledge, it is possible that in the future we will be able to modulate our microbiota by changing the “in-vironmental” milieu. This could be done by some nutritional interventions. This topic is now additionally discussed in our paper, according to the suggestions of the Reviewer, including the role of epigenetics. Nevertheless, in the light of current literature, we are not able to translate these data into any practical, nutritional recommendations.
Comment No 5: Conclusions are supported by insufficient evidence
Reply No 5: Dear Reviewer, according to you suggestion, we added a statement in the conclusions section that there are no data on the definite role of any diet in IBD. In light of this statement, we think that it is “reasonable” to promote dietitian support in elderly IBD patients, but this not a recommendation of a high-quality evidence.

Reviewer 3 Report
An interesting review on dietary support in elderly patients with IBD. With emphasis to practical guidelines. I have some comments and suggestions:
consider to add the mayor topics and conclusions to the abstract.
Pleas add the methods of the literature search. (which data bases; which key words etc)
line 60-61: Montreal classification: please add reference.
Line 67: study by, must be A study by.
Line 91: Ultrasound can be also advised. Better: ultrasound can also be advised.
Line 92-93: since repeated CT or MRI cn be difficult and in many cases contraindicated. Please describe why.
Line 107-110: Many of those conditions......significant drug interactions. Please add reference.
132-134: Etiology of this phenomenon....or growth hormone. Please add reference.
Line 147: It has meen also shown. Better: Also, it has been shown....
Line 152: Bowel wall destruction. the word "damage" seems more appropriate.
Line 181-182: Avoid high intake of fibre.... What is the lever of evidence? A reference is missing. In our opinion fibre can also have positive results in active IBD.
Line 201-202: Recently, also......has been discussed. Please add reference.
Line 204: Forbidden? or do you mean: advised not to use.
Line 211-214: The utility of low-FODMAP.....in genetically susceptible host. Please add reference.
Author Response
Dear Reviewer,
I would like to thank you for giving us a chance to revise our paper. The paper has been checked and corrected thoroughly according to all the comments. All corrections are marked in red in the file containing revised version of our paper.
Once again, I would like to express my sincere gratitude for giving us the opportunity to revise and resubmit our paper.
Reviewer No 3
An interesting review on dietary support in elderly patients with IBD. With emphasis to practical guidelines. I have some comments and suggestions:
Comment No 1: Coconsider to add the mayor topics and conclusions to the abstract.
Reply No 1: The major topics and conclusions are now more precisely defined in the abstract..
Comment No 2: Please add the methods of the literature search. (which data bases; which key words etc).
Reply No 2: We added the methodology of the literature search.
Comment No 3: Line 60-61: Montreal classification: please add reference.
Reply No 3: References have been added.
Comment No 4: Line 67: study by, must be A study by.
Reply No 4: Corrected as suggested by the Reviewer.
Comment No 5: Line 91: Ultrasound can be also advised. Better: ultrasound can also be advised.
Reply No 5: Corrected as suggested by the Reviewer.
Comment No 6: Line 92-93: since repeated CT or MRI cn be difficult and in many cases contraindicated. Please describe why.
Reply No 6: A significant proportion of older patients have renal insufficiency or are at risk of renal insufficiency, that is why repeated CT or MRI with IV contrast administration can be contraindicated. The frequency of different metallic implants (patients after total hip or knee replacement or after implantation of cardiac rhythm controlling devices) is also increased among older people, which is a contraindication to MR imaging. We have added these comments in our manuscript.
Comment No 7: Line 107-110: Many of those conditions......significant drug interactions. Please add reference.
Reply No 7: An additional reference has been added.
Comment No 8:132-134: Etiology of this phenomenon....or growth hormone. Please add reference.
Reply No 8: References have been added.
Reviewer No 9:Line 147: It has meen also shown. Better: Also, it has been shown....
Reply No 9: Corrected as suggested by the Reviewer.
Comment No 10:Line 152: Bowel wall destruction. the word "damage" seems more appropriate.
Reply No 10: Corrected as suggested by the Reviewer.
Comment No 11:Line 181-182: Avoid high intake of fibre.... What is the lever of evidence? A reference is missing. In our opinion fibre can also have positive results in active IBD.
Reply No 11: Dear Reviewer, we would like to thank you for this comment. This part of our paper refers to the exacerbation of IBD. We agree that the level of evidence is not high for this recommendation, however there are data and recommendations suggesting that low fiber intake can be helpful in patients with severe diarrhea or abdominal pain, especially in case of structuring CD. This recommendation does not apply to patients with UC and rectal involvement only who may develop constipation. That is why we use a term “It is advisable”, instead “it is recommended”.
We added some additional explanations in the text and additional reference.
Comment No 12:Line 201-202: Recently, also......has been discussed. Please add reference.
Reply No 12: References have been added.
Comment No 13:Line 204: Forbidden? or do you mean: advised not to use.
Reply No 13: We have changed this part of the sentence.
Comment No 14:Line 211-214: The utility of low-FODMAP.....in genetically susceptible host. Please add reference.
Reply No 14: References have been added.

Round 2
Reviewer 2 Report
In the paragraph where you discuss the gut microbiome - I think you should add some information about the impact of ageing on the microbiota - please review Paul O'toole's work from the ELDERMET cohort as I feel this relevant - the differences in the gut microbiome between community dwelling elderly and those living in long-term care facilities
In table 2 - regarding the Med Diet - where you state the Med diet consists of vegetable oils rich in unsaturated fatty acids, especially olive oil - I disagree - the focus of the Med diet is a diet rich in monounsaturated fats, such as olive oil NOT all unsaturated fatty acids - if you state unsaturated fatty acids then you include the fats rich in omega-6 when in fact this diet is low in omega-6 fatty acids - this needs to be corrected. Where you state that contraindication as vegetable oils rich in SFA - I am not sure this has been proven? Which fats do you mean? coconut oil? I agree with the contraindication as being animal fat - I think a better way to approach this is list out the specific fats versus lumping all fats together - as animal fat could mean dairy and there is some literature saying dairy has protective effects in IBD.
There should be a section in the paper discussing the impact of dietary fats, such as the impact of omega-6 on inflammation as this would support your argument as to why a Med diet could be considered in IBD.
There should be a section on dietary fibre again to support the argument of a Med diet.
Author Response
Dear Reviewer,
I would like to thank you for giving us a chance to revise our paper. The paper has been checked and corrected thoroughly according to all the comments. All corrections are marked in red in the file containing revised version of our paper.
Once again, I would like to express my sincere gratitude for giving us the opportunity to revise and resubmit our paper.
Reviewer No 2
Comment No 1: In the paragraph where you discuss the gut microbiome - I think you should add some information about the impact of ageing on the microbiota - please review Paul O'toole's work from the ELDERMET cohort as I feel this relevant - the differences in the gut microbiome between community dwelling elderly and those living in long-term care facilities
Reply No 1: Thank you for this comment. We added an additional paragraph about ELDERMET initiative and associations between ageing and changes in the microbiota composition.
Comment No 2: In table 2 - regarding the Med Diet - where you state the Med diet consists of vegetable oils rich in unsaturated fatty acids, especially olive oil - I disagree - the focus of the Med diet is a diet rich in monounsaturated fats, such as olive oil NOT all unsaturated fatty acids - if you state unsaturated fatty acids then you include the fats rich in omega-6 when in fact this diet is low in omega-6 fatty acids - this needs to be corrected. Where you state that contraindication as vegetable oils rich in SFA - I am not sure this has been proven? Which fats do you mean? coconut oil? I agree with the contraindication as being animal fat - I think a better way to approach this is list out the specific fats versus lumping all fats together - as animal fat could mean dairy and there is some literature saying dairy has protective effects in IBD.
Reply No 2: Thank you for your suggestions. The Table 2 has been corrected according to the comments of the Reviewer.
Comment No 3: There should be a section in the paper discussing the impact of dietary fats, such as the impact of omega-6 on inflammation as this would support your argument as to why a Med diet could be considered in IBD.
Reply No 3: Thank you for this comment. A separate caption has been added in which we discuss the role of PUFA in the context of possible modulation of inflammation in IBD.
Comment No 4: There should be a section on dietary fibre again to support the argument of a Med diet.
Reply No 4: Thank you for this comment. A separate caption has been added in which we discuss the role of dietary fiber in the context of possible modulation of inflammation in IBD.
